# Role of the YAP-1 Transcriptional Target cIAP2 in the Differential Susceptibility to Chemotherapy of Non-Small-Cell Lung Cancer (NSCLC) Patients with Tumor RASSF1A Gene Methylation from the Phase 3 IFCT-0002 Trial

**DOI:** 10.3390/cancers11121835

**Published:** 2019-11-21

**Authors:** Fatéméh Dubois, Maureen Keller, Julien Hoflack, Elodie Maille, Martine Antoine, Virginie Westeel, Emmanuel Bergot, Elisabeth Quoix, Armelle Lavolé, Laurence Bigay-Game, Jean-Louis Pujol, Alexandra Langlais, Franck Morin, Gérard Zalcman, Guénaëlle Levallet

**Affiliations:** 1Normandie Université, UNICAEN, CEA, CNRS, ISTCT/CERVOxy group, GIP CYCERON, 14074 Caen, France; fatemeh.dubois@unicaen.fr (F.D.); maureen.keller@ymail.com (M.K.); elo_61@hotmail.fr (E.M.); bergot-e@chu-caen.fr (E.B.); 2Department of Pathology, CHU de Caen, 14033 Caen, France; 3Normandie Université, UNICAEN, UPRES-EA2608, 14032 Caen, France; 4Normandie Université, UNICAEN, INSERM UMR 1086 ANTICIPE, 14032 Caen, France; 5Department of Pathology, Hôpital Tenon, AP-HP, 75020 Paris, France; martine.antoine@aphp.fr; 6Department of Pneumology, University Hospital of Besançon, University Bourgogne Franche-Comté, 25000 Besançon, France; virginie.westeel@univ-fcomte.fr; 7Department of Pulmonology & Thoracic Oncology, CHU de Caen, 14033 Caen, France; 8Department of Pneumology, University Hospital, 67000 Strasbourg, France; elisabeth.quoix@chru-strasbourg.fr; 9Sorbonne Université, GRC n 04, Theranoscan, AP-HP, Service de Pneumologie, Hôpital Tenon, 75020 Paris, France; armelle.lavole@aphp.fr; 10Pneumology Department, Toulouse-Purpan, University Hospital Toulouse, 31300 Toulouse, France; bigaygame.l@chu-toulouse.fr; 11Département d’Oncologie Thoracique, CHU Montpellier, Univ. Montpellier, 34595 Montpellier, France; jl-pujol@chu-montpellier.fr; 12Intergroupe Francophone de Cancérologie Thoracique (IFCT), 75009 Paris, France; alexandra.langlais@ifct.fr (A.L.); franck.morin@ifct.fr (F.M.); 13U830 INSERM “Genetics and Biology of Cancers, A.R.T Group”, Curie Institute, 75005 Paris, France; 14Department of Thoracic Oncology & CIC1425, Hôpital Bichat-Claude Bernard, Assistance Publique Hôpitaux de Paris, Université Paris-Diderot, 75018 Paris, France

**Keywords:** non-small-cell lung cancer, RASSF1A, DNA methylation, paclitaxel, IAP2

## Abstract

*RASSF1* gene methylation predicts longer disease-free survival (DFS) and overall survival (OS) in patients with early-stage non-small-cell lung cancer treated using paclitaxel-based neo-adjuvant chemotherapy compared to patients receiving a gemcitabine-based regimen, according to the randomized Phase 3 IFCT (Intergroupe Francophone de Cancérologie Thoracique)-0002 trial. To better understand these results, this study used four human bronchial epithelial cell (HBEC) models (HBEC-3, HBEC-3-RasV12, A549, and H1299) and modulated the expression of RASSF1A or YAP-1. Wound-healing, invasion, proliferation and apoptosis assays were then carried out and the expression of YAP-1 transcriptional targets was quantified using a quantitative polymerase chain reaction. This study reports herein that gemcitabine synergizes with RASSF1A, silencing to increase the IAP-2 expression, which in turn not only interferes with cell proliferation but also promotes cell migration. This contributes to the aggressive behavior of RASSF1A-depleted cells, as confirmed by a combined knockdown of IAP-2 and RASSF1A. Conversely, paclitaxel does not increase the IAP-2 expression but limits the invasiveness of RASSF1A-depleted cells, presumably by rescuing microtubule stabilization. Overall, these data provide a functional insight that supports the prognostic value of *RASSF1* gene methylation on survival of early-stage lung cancer patients receiving perioperative paclitaxel-based treatment compared to gemcitabine-based treatment, identifying IAP-2 as a novel biomarker indicative of YAP-1-mediated modulation of chemo-sensitivity in lung cancer.

## 1. Introduction

Over the last few years, targeted therapy based on specific predictive biomarkers has improved both the survival rate and quality of life of cancer patients [1,2]. Among the predictive biomarkers in patients with non-small-cell lung cancer (NSCLC), the promoter hyper-methylation of the tumor suppressor gene *RASSF1A* is still misused. However, the results of the Phase 3 IFCT (Intergroupe Francophone de Cancérologie Thoracique)-0002 randomized trial demonstrated both the prognostic and predictive values of *RASSF1A* gene silencing, following neo-adjuvant chemotherapy in patients with Stage I–II NSCLC [3]. The patients with *RASSF1A* promoter gene methylation displayed a three-fold decrease in the 5-year overall survival (OS) rate [3]. Additionally, a worse median OS was observed in patients with methylated *RASSF1A* treated with gemcitabine (30.3 months) compared to those treated with paclitaxel (70 months) [3]. These prognostic values of *RASSF1A* gene methylation were supported by data that demonstrated that RASSF1A restricts epithelial-mesenchymal transition (EMT) and cell invasion by controlling Yes-associated protein (YAP) nuclear shuttling and RhoB-regulated cytoskeletal remodeling process [4,5]. As such, RASSF1A inactivation favors the acquisition of a metastatic phenotype that explains these patients. However, how RASSF1A epigenetic silencing contributes to the positive effects of paclitaxel versus gemcitabine treatment has yet to be determined [3]. To be able to rationally develop enhanced treatment strategies, it is imperative to define whether RASSF1A depletion enhances sensibility to paclitaxel or, to the contrary, increases the patient’s resistance to gemcitabine-induced cell death. 

Paclitaxel is a tubulin-stabilizing agent that leads to mitotic arrest, while gemcitabine is a cytosine analogue that inhibits nucleoside metabolism, both ultimately causing cell death [6,7]. Both drugs have become key components in the treatment of advanced NSCLC patients, being given mostly in combination with platinum compounds [8,9] prior to the introduction of immune checkpoint inhibitors (ICI) for managing Stage IV NSCLC patients. This triple combination (platinum-based chemotherapy and ICI) is being currently tested in a neo-adjuvant setting. Based on post-hoc biomarker analyses of clinical trials, the predominant hypothesis explaining such data would be that paclitaxel mimics *RASSF1A*-induced microtubule stabilization. However, numerous studies have also demonstrated the relevance of RASSF1A’s association with microtubules and members of the Hippo pathway in the apoptosis process [10,11,12]. As such, RASSF1A depletion may lead to the cells’ resistance to apoptosis, along with the subsequent ineffectiveness of chemotherapeutic agents. This resistance accounts for difficulties in the management of numerous cancers, due to either cellular adaptation to stress or an increase in anti-apoptotic proteins, such as the YAP-1 transcriptional target inhibitor of apoptosis protein (IAP) [13]. Indeed, through their three baculovirus IAP repeat (BIR) domains, IAPs bind to and inhibit caspases, which act as the terminal effectors of apoptosis [14,15]. However, IAPs represent a highly versatile class of proteins that regulate not only apoptosis but also various biological functions, including innate immunity, inflammation, cell proliferation and cell migration [16,17,18,19,20]. Some members of the IAP family, such as IAP-1 and IAP-2, have been reported to be overexpressed in several cancers, including lung cancer [21,22,23,24].

As some IAP family members were reported to be transcriptional targets of Yorkie, YAP drosophila ortholog [13], the activity of which is increased and deregulated in RASSF1A-depleted NSCLC cells [4], this study hypothesized that IAPs could underlie the predictive value of RASSF1A promoter methylation. By using either RNAi to mimic RASSF1A methylation in NSCLC or RASSF1A-encoding plasmid in rescue experiments, this study provided molecular insights into the prognostic value of RASSF1A and how its depletion would possibly affect the efficiency of chemotherapeutic agents like paclitaxel and gemcitabine. Moreover, IAP-2 as a novel putative biomarker was introduced for choosing paclitaxel rather than gemcitabine for the platinum-based doublets, which are still used as perioperative treatments in patients with early-stage NSCLC and silenced RASSF1A.

## 2. Results

### 2.1. RASSF1A Depletion Suppresses Cell Sensitivity to Drug-Induced Apoptosis

RASSF1A’s role was first investigated in modulating apoptosis in response to chemotherapeutic agents, such as paclitaxel and gemcitabine, using non tumorigenic and untransformed but immortalized bronchial HBEC-3 and HBEC-3 RasV12 cell lines [25], in order to mimic the early loss of RASSF1A expression in lung epithelial carcinogenesis [26]. RASSF1A was depleted using two different small interfering RNAs (siRNA), leading to 70% and 80% decreases in RASSF1A mRNA and protein levels, respectively (HBEC-3: Appendix A; HBEC-3 RasV12: Appendix A), as previously reported [4]. The cancer-derived A549 and H1299 cell lines with *RASSF1A* promoter gene methylation were additionally used and no basal RASSF1A protein expression in rescue experiments in order to confirm the specificity of our RNA-interference (RNAi) results. Accordingly, RASSF1A was reintroduced using a RASSF1A-encoding expression plasmid (H1299: Appendix A; A549: Appendix A).

Twenty-four hours after being transfected with the constructs (control RNAi [siNeg], siRASSF1A-1 or -2, control [Pls Ctr] and RASSF1A-encoding plasmids [Pls RASSF1A]), the cells were treated with either paclitaxel (10 nM) or gemcitabine (250 nM) for another 24 h (Figure 1). Etoposide (50 µM) was employed as an apoptosis inducer and a positive control for drug efficacy [27]. As expected, the control cells’ (siNeg or Pls Ctr) exposure to either paclitaxel or gemcitabine caused a significant increase in caspase 3/7 activities, cytochrome c release and DNA fragmentation after the cells were treated with chemotherapy (HBEC-3: Figure 1A,C,D; HBEC-3 RasV12: Figure 1B–E; H1299: Appendix A; and A549: Appendix A, respectively). With the exception of A549 cells, in our experimental conditions, paclitaxel was more likely to induce apoptosis than gemcitabine (HBEC-3: Figure 1A,C,D; HBEC-3 RasV12: Figure 1B–E; H1299: Appendix A; and A549: Appendix A).

Following RASSF1A knockdown in untreated HBEC-3 and HBEC-3 RasV12 cell lines, this study did not observe any significant differences in caspase 3/7 activities (HBEC-3: Figure 1A; HBEC-3 RasV12: Figure 1B) or DNA fragmentation (HBEC-3: Figure 1D; HBEC-3 RasV12: Figure 1E). However, surprisingly, RASSF1A silencing dramatically reduced the effect of paclitaxel treatment on both caspase 3/7 activities and DNA fragmentation (HBEC-3: Figure 1A–D; HBEC-3 RasV12: Figure 1B–E).

Consistently, the exogenous expression of RASSF1A in H1299 and A549 cells, which display no detectable basal expression of RASSF1A, did not influence caspase3/7 activities (H1299: Appendix A; A549: Appendix A). However, paclitaxel either tended to increase (A549) or increase (H1299) the efficiency of caspase 3/7 activities following the reintroduction of RASSF1A (H1299: Appendix A; A549: Appendix A).

Altogether, these data indicate that RASSF1A knockdown reduces cell sensitivity to drug-induced apoptosis.

### 2.2. RASSF1A Depletion Coincides with Strong YAP-Dependent IAP-2 Expression

The authors have previously demonstrated that the loss of RASSF1A expression leads to the inappropriate activity of YAP, the subcellular localization of which becomes preferentially nuclear and coincides with an increase in the transcription of several target genes [4,5,28]. Some IAPs, proteins acting as negative apoptosis regulators by inhibiting the activity of several caspases [29], could also be the transcriptional targets of YAP as shown in drosophila [13]. The impact of RASSF1A levels was then tested on the expression of different IAPs (Figure 2). It was found that RASSF1A-depleted HBEC-3 cells exhibited a significant increase in IAP-2 mRNA expression, while there were no changes in the IAP-1, XIAP, or Bruce mRNA levels (Figure 2A). Conversely, the exogenous expression of RASSF1A in A549 and H1299 cells significantly reduced the IAP-2 mRNA expression levels compared to those of the control cells transfected with a mimic plasmid (H1299: Figure 2B; A549: Figure 2C). Here, the absence or presence of RASSF1A again had no effect on the other apoptosis inhibitor levels (H1299: Figure 2B; A549: Figure 2C). Next, by co-transfection of the HBEC-3 cells with YAP and RASSF1A RNAi, it was further confirmed that the YAP transcriptional activity was responsible for IAP-2 expression in RASSF1A-depleted cells (Figure 2D).

It was thus assumed that RASSF1A depletion leading to YAP activation could decrease cancer cell apoptosis by increasing the expression of the inhibitor of apoptosis protein IAP-2.

### 2.3. Strong YAP Intensity Associated with a Weak Response Rate to Chemotherapy in NSCLC Patients from the IFCT-0002 Trial

To investigate whether YAP expression is involved in NSCLC patients’ responses to chemotherapy, the status and subcellular localization of YAP were evaluated in the available samples from 528 patients included into the Phase III clinical Bio-IFCT0002 cohort [3] (Appendix A). The sections of formalin-fixed, paraffin-embedded tissue samples were immunostained using a YAP antibody, with the YAP expression determined via an H-score method (intensity score (Appendix A) × percentage of positive tumoral cells). The analyses of 362 out of the 443 available tumor samples were informative for YAP immunostaining (Appendix A). Overall, 43 tumor samples (11.9%) were YAP negative, 245 (67.6%) had mild to moderate YAP intensity and 74 (20.4%) had high YAP intensity. The subcellular localization of YAP occurred exclusively in the cytoplasm in 21.3% of cases, exclusively in the nucleus in 51.7% and across the cytoplasm and nucleus in 27% of cases. The median immunostaining H-score was 80.

The univariate analysis revealed no prognostic impact of YAP intensity on NSCLC patient survival (Appendix A), while the localization of YAP within the cytoplasm, where YAP is sequestered or prone to proteasome degradation, tended to predict longer OS. However, without reaching statistical significance (Appendix A, *p* = 0.24, median survival of ~65 months versus >110 months for patients with nuclear YAP), in addition to longer progression-free survivals (PFS) of these patients (*p* = 0.2, median survival of ~42 months versus 32 months for patients with nuclear YAP). As the authors suspected a lack of statistical power due to the attrition of the numbers of available pathological samples from the IFCT-0002 cohort, the survival data was further analyzed following resections from early-stage lung cancer patients from the publicly available Cancer Tumor Genome Atlas (CTGA) cohort. This analysis revealed that a high mRNA expression was associated with worse OS rates of NSCLC patients (Appendix A).

Finally, YAP intensity was reported to be associated with objective responses (partial or complete) of early-stage NSCLC patients treated with preoperative chemotherapy in the IFCT-0002 trial (Table 1): Overall, 63 of the 356 (17.7%) patients studied in this trial showed no response and did not express or expressed low (0–1 intensities) tumoral YAP levels (37.6% [63/134] of patients with low YAP intensity tumor), while 135 patients showed no response and a strongly expressed YAP tumor intensity (60.81% [135/222] of patients with strong tumoral intensity of YAP) (chi2 *p*-value = 0.011).

Therefore, it was postulated that the RASSF1A depletion leading to YAP activation likely decreases cancer cells’ apoptosis by increasing the expression of the inhibitor of the apoptosis IAP-2 protein. This rationale supports the observation that strong YAP expression was associated with a high progression rate in NSCLC patients from the IFCT-0002 trial.

### 2.4. Gemcitabine Treatment Synergizing the Increased IAP-2 Expression Induced by RASSF1A Depletion

Given that several commonly used antineoplastic drugs are able to increase IAPs’ expression [29,30], this study then evaluated the impact of paclitaxel and gemcitabine treatments on the IAP-2 levels present in or absent from *RASSF1A* gene expression. It was observed that gemcitabine treatment alone caused a significant increase in IAP-2 expression in HBEC-3 control cells. In combination with RASSF1A RNAi depletion, gemcitabine treatment exerted a cumulative effect (Figure 3A). In contrast, the co-treatment by paclitaxel and RASSF1A RNAi knockdown on IAP-2 expression was comparable in their effects to that of RASSF1A-depleted cells alone (Figure 3A).

Consistently, treating the H1299 cells with the methylated *RASSF1A* gene using gemcitabine alone significantly increased IAP-2 mRNA levels (Figure 3B), while the authors did not observe any IAP-2 mRNA expression changes in A549 *RASSF1A* methylated cells treated with either paclitaxel or gemcitabine (Figure 3C). Furthermore, whereas the ectopic expression of RASSF1A significantly reduced IAP-2 mRNA levels in both lines, the simultaneous cell transfection with RASSF1A plasmid and gemcitabine treatment had an additive effect on the reduced IAP-2 expression (H1299: Figure 3B; A549: Figure 3C). Intriguingly, the transfection of the H1299 and A549 cells with RASSF1A plasmid along with paclitaxel treatment did not change the IAP-2 levels compared to either the control or paclitaxel-treated cells alone.

Therefore, this study postulated that gemcitabine treatment specifically synergizes the IAP-2 expression increase induced by RASSF1A depletion.

### 2.5. IAP-2 Interfering with RASSF1A-Mediated Cell Proliferation

Furthermore, IAPs prevent apoptosis by inhibiting caspase activation, thereby interfering with cell proliferation and survival [18,31]. To investigate whether RASSF1A-dependent modulation of IAP expression could affect proliferation and apoptosis, the cells were transfected with different combinations of RASSF1A constructs and IAP-1 and two siRNAs. Both RNAi resulted in the efficient decrease in IAP-1 or IAP-2 mRNA expression by 80–90% in HBEC-3 (Appendix A), H1299 (Appendix A) and A549 (Appendix A) cell lines.

The cells first underwent a proliferation analysis using bromodeoxyuridine (BrdU) incorporation assays. As shown in Figure 4A, HBEC-3 cell proliferation significantly decreased following RASSF1A depletion, whereas neither siIAP-1 nor siIAP-2 alone had any influence on cell proliferation (Figure 4A). Remarkably, the concomitant RASSF1A and IAP-2 depletion was shown to rescue the RASSF1A-depleted cells’ proliferation defect (Figure 4A). In contrast, a combined knockdown of RASSF1A and IAP-1 RNAi had no impact on the reduced cell proliferation induced by RASSF1A depletion in HBEC-3 cells (Figure 4A).

Moreover, exogenous RASSF1A expression caused a slight decrease in cell proliferation in H1299 cells (Figure 4B) and A549 cells (Figure 4C), the difference achieved being statistically significant only in A549 cells (Figure 4C). This reflects the necessity of appropriate RASSF1A levels for efficient cell proliferation [5,32]. To the authors’ surprise, in both H1299 and A549 cells, a single IAP-2 knockdown or a combined IAP-2 knockdown with RASSF1A re-expression caused a modest but significant reduction in cell proliferation (Figure 4B,C). This inhibitory effect on proliferation was also observed in A549 cells following the transfection of IAP-1 RNAi and RASSF1A plasmids (Figure 4C). Hence, it was postulated that IAP-2 could interfere with RASSF1A-mediated cell proliferation.

### 2.6. Increased IAP-2 Expression both Necessary and Sufficient for siRASSF1A-Induced Increase in Cell Migration and Invasion

The increased migration and invasiveness are striking hallmarks of cancer cells [33]. A body of evidence supports IAPs to play a critical role in regulating cell migration and metastasis [34,35]. Since RASSF1A controls both cell migration and invasion [4,36], IAPs’ involvement in these processes was next explored. Consistent with our previous experiments, a wound-healing assay revealed the increased migration of RASSF1A-depleted HBEC-3 cells compared to that of the controls (Figure 5A), whereas RASSF1A re-expression was associated with a significant reduction in migration distances of H1299 and A549 cells lines (Figure 5C and Appendix A). Further results in HBEC-3 cells revealed that, compared to the depletion of either IAP-1 or IAP-2 alone exhibiting no effect on migration velocity in control cells (siNeg) with normal RASSF1A expression, a combined knockdown of IAP-1 and RASSF1A or IAP-2 and RASSF1A significantly decreased the siRASSF1A-induced enhancement of cell migration (Figure 5A). Accordingly, IAP-1 silencing did not affect the migration velocity of H1299 and A549 cells either expressing or not RASSF1A, whereas IAP-2 silencing significantly decreased such migration velocity of H1299 and A549 cells either expressing or not (Figure 5C and Appendix A). These data prompted us to postulate that increased IAP-2 expression following RASSF1A depletion could be necessary for the observed increment in cell migration.

Next, the Matrigel^®^-coated Transwell 3D migration was exploited. As expected, based on our previous results, RASSF1A depletion effectively promoted cell invasion through Matrigel^®^ in HBEC-3 cells (Figure 5B). In contrast, exogenous RASSF1A expression significantly reduced H1299 and A549 cells’ 3D migration and invasion through Matrigel^®^ (Figure 5D and Appendix A). Remarkably, IAP-1 depletion alone was accompanied by a similar increase in HBEC-3 cell invasiveness. However, a combined RASSF1A and IAP-2 depletion dramatically blocked the siRASSF1A-induced invasion (Figure 5B). Here, the elimination of endogenous IAP-2 from both H1299 and A549 cells lines caused a reduction in the invasive capacity of either the control or RASSF1A re-expressed cells to the level observed in cells transfected with exogenous RASSF1A (Figure 5D and Appendix A), whereas IAP-1 depletion only affected the 3D migration of H1299 cells (Figure 5D).

Considering the IAP-2 reduction in cells transfected with RASSF1A plasmid (Figure 2B,C), these data support IAP-2 to possibly represent a critical downstream target of RASSF1A for controlling cell migration and invasion.

### 2.7. Paclitaxel Treatment Rescuing Normal Invasion Following RASSF1A Knockdown

RASSF1A controls cell migration and invasion by its association with microtubules, resulting in their stabilization [37,38]. Paclitaxel also binds reversibly to microtubules, and the resulting microtubules are stable and resistant to tubulin depolymerization [39]. Given that RASSF1A depletion induces microtubule destabilization and therefore increases cell migration and invasion [4], it was queried whether paclitaxel could rescue these effects if microtubule stability increased. To explore this possibility, this study compared the effect of paclitaxel and gemcitabine alone or in combination with either siRASSF1A or plasmids encoding wild-type RASSF1A on cell invasion ability.

In agreement with previous data [4], a significant increase in the cell invasion ability of HBEC-3 RASSF1A-depleted cells were found, whereas, conversely, RASSF1A overexpression following HBEC-3 cell transfection of an RASSF1A-encoding plasmid caused a significant decrease in cell invasion (Figure 6). Remarkably, compared to untreated controls and gemcitabine-treated cells, paclitaxel exposure was found to be able to restore basal invasion capacities in RASSF1A-knockdown cells (Figure 6). Based on these results, normal microtubule dynamics per se proved to be sufficient for the efficient invasion properties of cells.

## 3. Discussion

The authors were intrigued to further understand how the RASSF1A expression loss could predict either the effectiveness of pre-operative taxol-based therapy or the failure of pre-operative gemcitabine treatment in patients with early-stage NSCLC from the Phase 3 IFCT-0002 trial, as reported in detail in our previous work [3]. Herein, some evidence was provided supporting that gemcitabine synergizes with the RASSF1A silencing in human epithelial immortalized bronchial cells (HBEC) in order to increase IAP-2 expression. This latter, in turn, interferes with cell proliferation, promotes cell migration and thus contributes to the aggressive behavior of such RASSF1A-depleted cells. Conversely, paclitaxel does not interfere with IAP-2 expression, and thus contributes efficiently to limit cell migration and subsequent metastatic potential, which accounts for the survival effect of *RASSF1A* promoter gene methylation in paclitaxel-treated NSCLC patients (Graphical abstract).

A major mode of action of chemotherapeutic drugs is to provoke apoptosis [40], whereas RASSF1A is a well-established regulator of apoptosis [41]. Thus, the authors first decided to evaluate the activity of the known downstream proteins of the apoptotic pathway in HBEC, with and without RASSF1A expression [42]. This study chose to use the non tumorigenic and untransformed (yet immortalized) bronchial HBEC-3 and HBEC-3 RasV12 cell lines [25] so as to mimic the early-stage loss of RASSF1A expression in lung pre-neoplastic epithelium [26], as well as in cancer-derived RASSF1A-methylated A549 and H1299 cell lines for rescue experiments by re-expressing RASSF1A in such cells. While the loss-of-function of RASSF1A intervenes during early carcinogenesis stages, most of the data concerning RASSF1A’s role in apoptosis were derived from highly invasive cancer cells [43,44,45,46]. Using a clinically relevant dose of each drug [47], this study reported that paclitaxel-treated HBEC displayed more increased apoptosis than gemcitabine-treated HBEC (Figure 1 and Appendix A, respectively). The differences in response patterns between the two drugs could be attributed to their ability to regulate specific components of cell death pathways. Paclitaxel-induced apoptosis is mainly mediated through the enforced activation of various caspases, including caspase 3 and 7 [48,49,50]. Gemcitabine primarily causes Poly (ADP-ribose) polymerase (PARP) degradation through autophagy-regulated processes rather than apoptosis [51], which is why the results of single-agent gemcitabine chemotherapy are significantly improved by adding a PARP inhibitor in BRCA-1-deficient pancreatic and breast cancer cell lines [52,53,54].

The present results demonstrated that the knockdown of RASSF1A reduced the sensitivity of HBEC-3 cells to apoptosis after being treated with either paclitaxel or gemcitabine (Figure 1). In contrast, RASSF1A re-expression exerted the opposite impact, thereby promoting gemcitabine-mediated apoptosis induction in RASSF1A-deficient H1299 cells (Appendix A). These functional studies support the anti-apoptotic role of RASSF1A [41]. Our work confirms the findings concerning an acquired resistance to paclitaxel after RASSF1A depletion (already observed in primary ovarian cancer cell models) [55], while elucidating how the RASSF1A loss could lead to a decreased gemcitabine effectiveness.

Next, very surprisingly, RASSF1A was shown to be able to modulate IAP-2 expression in a YAP-dependent manner (Figure 2). This outcome appeared to be quite specific, given that this study did not observe any significant expression changes in other IAP family members (IAP-1, survivin, or Bruce) through the modulation of RASSF1A expression (Figure 2) despite the fact that survivin expression has been previously reported to be increased in RASSF1A-depleted SKOV-3 ovarian cancer cells [46]. These data are consistent with the previous description of some IAPs as transcriptional targets of the transcriptional cofactor YAP-1 [13]. This latter result was previously reported by the authors to accumulate in the nucleus with transcriptional activation after RASSF1A knockdown [4]. Further, IAPs are known to be negative regulators employed by cancer cells in order to suppress apoptosis in numerous histological cancer subtypes. Of note, this suppression of apoptosis plays a central role upon multi-step carcinogenesis [13,56]. Interestingly, the overexpression of both IAP and YAP-1 genes was reported to exert a synergistic effect in promoting tumor-genesis [57]. Accordingly, using the samples collected from the Phase 3 IFCT-0002 trial patients [3], it was found that the YAP-1 nuclear localization in such samples was associated with a numerical, although not significant, decrease in OS (Appendix A). Most importantly, such strong YAP expression was associated with a higher patients’ unresponsiveness rate to preoperative chemotherapy (Table 1). Unfortunately, this study could not quantify the IAP-2 expression in these same patient samples, owing to of a lack of reliable antibodies available for immunohistochemistry application of formalin-fixed paraffin-embedded (FFPE) samples. Furthermore, this sample collection is currently not in use, given that this specific analysis was not pre-specified or initially planned.

Additionally, a cumulative effect was observed of either RASSF1A knockdown or RASSF1A re-expression in combination with gemcitabine treatment, resulting in a dramatic increase or decrease in IAP-2 expression, respectively (Figure 3 and Appendix A). As RASSF1A has been clearly identified as a tumor suppressor gene [41], whereas IAPs are supposed to act as oncogenes [13], the authors suspect that this increase in IAP-2 could contribute to the impact of RASSF1A expression loss on tumor promotion. In support of this, the combined knockdown of RASSF1A and IAP-2 was able to restore the number of proliferative cells to the control cell level (Figure 4). Remarkably, either IAP-2 knockdown alone or IAP-2 knockdown associated with RASSF1A forced re-expression and modestly inhibited cell proliferation. Of note, this observation highlights the context-dependent role IAPs play in controlling cell proliferation. For instance, it has been demonstrated that nuclear BIRC5/survivin, another IAP family member, regulates proliferation, whereas the cytoplasmic survivin protein pool acts to suppress apoptosis [58]. Published data reported that IAPs’ increased expression following gemcitabine treatment is able to suppress apoptosis, thus resulting in chemotherapy resistance [29,59]. Our data support that IAP-2 overexpression upon gemcitabine treatment likely proved to be the primary mechanism that counterbalances apoptotic signaling in HBEC-3 cells.

Our further mechanistic studies demonstrated that not only did the depletion of both RASSF1A and IAP-2 decrease the migration and invasion ability of RASSF1A-depleted cells, but the increase in IAP-2 expression also seemed to be critical in the control of both migration and invasion through RASSF1A (Figure 5 and Appendix A). These data are in concordance with previous reports demonstrating that IAPs are deemed crucial regulators of tumor cell migration and metastasis [34]. For example, IAP-1 depletion has been reported to be able to suppress the Matrigel^®^ invasion of PC3 prostate cancer cells [60]. Collectively, these data underscore the pertinence of IAP-2 overexpression in mediating the effects of RASSF1A down-regulation. All this suggests that IAP-2 could be a critical downstream target of RASSF1A for controlling cell migration and invasion, although mechanistic studies are still pending to fully elucidate the mechanisms involved in such functional interactions.

Besides its general role of an apoptosis regulator [61], RASSF1A is known to induce microtubule stabilization [38,62]. The inactivation of RASSF1A thus results in an increased sensitivity to microtubule-destabilizing drugs [37], and RASSF1A overexpression is surprisingly reminiscent of the cell effects produced by paclitaxel [63]. This study therefore, hypothesized that paclitaxel-induced microtubule stabilization may likely be able to rescue the effects of RASSF1A depletion. The invasion assay was used to test such a hypothesis. Accordingly, compared to the untreated control and gemcitabine-treated cells, paclitaxel exposure was able to restore normal invasion in RASSF1A knockdown cells (Figure 6). These results corroborate previous study data that revealed a reduction in cell invasiveness following paclitaxel treatment administered at sub-lethal doses [64,65,66]. Conversely, other authors reported the opposite effect [67]. Most of the discrepancies pertaining to the pro- and anti-migratory effects of paclitaxel observed could be attributed not only to the differences in doses and cell subtypes employed, but also to the basic fact that appropriate microtubule dynamics are necessary to maintain normal cellular function. Since our data suggest a positive role of paclitaxel through microtubule stabilization, it is conceivable that microtubules might be indirectly involved in apoptosis regulation by recruiting, transporting, or both, the implicated proteins. Thus far, the interaction of the survivin protein with microtubules has been attributed to an increase in MT stability, which resulted in improved clinical outcomes [68]. Whether a similar function applies to other IAPs remains to be explored.

## 4. Materials and Methods

### 4.1. Patients and Bio-IFCT 0002 Trial

Between 2001 and 2005, 528 patients were recruited into the Phase 3 IFCT 0002 trial, which was approved by the corresponding ethics committee (CPPRB of University Hospital in Besançon, France, ethic code: 00/282). Specific informed consent deigned for biological studies was obtained (Bio-IFCT 0002) beforehand. Two platin-based perioperative chemotherapy regimens, gemcitabine plus cisplatin and paclitaxel plus carboplatin, and two chemotherapy schedules were compared in patients with resectable Stage I or II NSCLC. In the preoperative arm (PRE), the patients received two courses of either chemotherapy regimen. The patients who did not respond discontinued chemotherapy and underwent surgical resection, whereas the patients with a partial response received two more cycles prior to surgical resection. In the perioperative arm (PERI), the patients received two courses of either chemotherapy regimen and underwent surgical resection. Only responder patients received two additional adjuvant cycles. The results of this Phase 3 trial have been presented elsewhere [69]. A Bio-IFCT 0002 study was designed by a steering committee, conducted according to a detailed protocol, and granted by the *Programme Hospitalier de Recherche Clinique* (PHRC) national 2001. 

### 4.2. Cell Culture, siRNA, Constructs, Transfection, and Treatments

Immortalized human bronchial epithelial HBEC-3 and HBEC-3 RasV12 cells were kindly provided by Dr. Michaël White (UT Southwestern Medical Center, Dallas, TX, USA), as previously described [70]. They were grown in keratinocyte serum-free medium (KFSM) supplemented with 0.2 ng/mL epidermal growth factor (EGFr) and 25 μg/mL bovine pituitary extracts (BPE) (Thermo Fisher Scientific, Rockford, IL, USA). Cancer-derived cell lines A549 and H1299 were purchased from the American Type Culture Collection (ATCC) and were grown in Dulbecco’s Modified Eagle Medium (DMEM) supplemented with 10% (*vol/vol*) heat-inactivated fetal bovine serum. Both KSFM and DMEM mediums were complemented by 100 U/mL penicillin, 100 μg/mL streptomycin and 2 mM l-glutamine (Gibco, Life Technologies, Grand Island, NY, USA). The cultures were incubated at 37 °C in a humidified atmosphere with 5% CO_2_. Where indicated, the cells were treated with the appropriate drug concentration (either 10 nM paclitaxel or 250 nM gemcitabine) for 24 h before analysis was conducted based on the previous study [47]. The 50 µM etoposide was employed as a positive control [27]. These drugs were purchased from Selleck Chemicals Co., Ltd. (Houston, TX, USA).

The following RNAi oligonucleotides from Eurogentec^®^ were employed: RASSF1A: 5′-GACCUCUGUGGCGACUUCA(TT)-3′ and 5′-GAACGUGG ACGAGCCUGU(TT)-3′ [4]; to deplete IAP-1 expression, this study used 5′-GAAUGAAAGGCCAAGAGUU(TT)-3′ and 5′-CAGAAAGCUUUGAAUACUA(TT)-3′ [71] and for IAP-2 5′-AAUGCAGAGUCAUCAAUUA(TT)-3′ [72] and 5′-AAUGAUGGUUGAAGGUUACAU (TT)-3′ [73]. The mock siRNA was employed as the non-silencing negative control (Dharmacon, Thermo Scientific, Pittsburgh, PA, USA). Plasmids encoding wild-type RASSF1A (pcDNA3-RASSF1A) and control mimic (Addgene, Cambridge, MA, USA) have been described in the supplementary data section [4]. The introduction of siRNA and plasmids was performed using Lipofectamine RNAiMax (Invitrogen, Carlsbad, CA, USA) in accordance with the manufacturer’s instructions at 30% and 70% of cell confluence, respectively.

### 4.3. Preparation of RNA and RT-PCR

The extraction of total RNA from treated and untreated cells was carried out using the illustra RNAspin mini^®^ column (GE Healthcare, Bio-Sciences, Pittsburgh, PA, USA), according to the manufacturer’s instructions. Total RNA was treated with DNAse I (Invitrogen, Carlsbad, CA, USA) in order to remove contaminating genomic DNA. The RNA concentrations were determined using spectrophotometer Nanodrop^®^ 2000c. The total RNA (250 ng) was reverse-transcribed with random primers and 200 IU M-MLV reverse transcriptase at 37 °C for 90 min, followed by 5 min of dissociation at 70 °C with Mastercycler Eppendorf^®^. The resulting cDNAs were diluted (1/10) and used as templates. Polymerase chain reaction (PCR) was performed in a Mx3005P QPCR system (Agilent Technology) with 5pmol of each primer set (Table 2) and iQTM SYBR Green Supermix (Bio-Rad, Hercules, CA, USA) as follows: 95 °C for 5 min, followed by 40 cycles at 95 °C for 1 min, and annealing/extension at 60 °C for 60 s. Furthermore, S16 was used as an internal control. Positive standards and reaction mixtures lacking the reverse transcriptase were employed routinely as controls for each RNA sample. The relative quantification was calculated using the ΔΔCt method.

### 4.4. Antibodies

The following antibodies were used: monoclonal mouse anti-human Cytochrome c (BD Transduction Laboratories, San Jose, CA, USA), monoclonal mouse anti-human RASSF1A (Ebioscience, San Diego, CA, USA) and monoclonal rabbit anti-human PARP (Cell Signaling Technology, Beverly, MA, USA).

### 4.5. Immunofluorescence (IF), Immunohistochemistry (IHC), and Image Analysis

For IF studies, the cells were seeded on coverslips in 24-well tissue culture trays at a density of 2 × 10^4^. Following 48 h, the cells were washed with PBS and fixed with 4% paraformaldehyde for 20 min at 37 °C. The cells were then permeabilized with frozen methanol for 10 min and blocked with 4% bovine serum albumin (BSA) in phosphate-buffered saline (PBS) for 1 h and stained with primary antibodies at 4 °C overnight. After being washed with PBS, the cells were stained with either Alexa-488-conjugated or Alexa-555-conjugated secondary antibodies (Molecular Probes, Invitrogen, Eugene, OR, USA) for 1 h at room temperature (RT) and with DAPI (4,6 diamidino-2-phenylindole) (Santa Cruz Biotechnology, Dallas, TX, USA). Digital pictures were captured using a high-throughput confocal microscopy (FluoView FV1000, Olympus, Wendenstrasse, Germany).

The IHC was performed, as described previously [70,74]. Briefly, 443 out of 492 paraffin-embedded blocks issued from the Phase 3 IFCT-002 trial were collected for this study. A YAP antibody was used at 1:100 dilution. The internal positive controls were systematically performed for each tumor series (immuno-stained basal cells). The staining intensities were evaluated in a blinded manner at 40× magnification and were scored using marker-specific 0–3 scales (0: negative, 1: weak, 2: moderate, and 3: strong). An overall IHC composite score was calculated based on the sum of the staining intensity (0–3) multiplied by the distribution (0–100%) from all parts of the slide, thereby providing an H-score between 0 and 300.

### 4.6. BrdU Incorporation Assay

The effect of chemotherapy drugs on cell growth was evaluated using a BrdU incorporation assay kit in accordance with the manufacturer’s instructions (1/500 dilution) (cat. no. 2750, Millipore, Billerica, MA, USA). BrdU, a synthetic thymidine analogue, can be incorporated into newly synthesized DNA, providing a test of DNA replication as an indirect measure of cell proliferation. The BrdU incorporation was detected by adding a peroxidase substrate. Spectrophotometric detection was performed at 450 nm wavelength.

### 4.7. Apoptosis Measurement

DNA fragmentation and caspase-3/7 activation were assayed using the Cell Death Detection ELISA Plus Kit (Roche Diagnostics, Mannheim, Germany) and the Caspase-Glo 3/7 Luminescence Assay (Promega Corp., Madison, WI, USA), respectively, according to the manufacturer’s instructions.

### 4.8. Wound-Healing Assay

The transfected cells were seeded on collagen IV-coated plates (BD BioCoat Matrigel^®^ Invasion Chamber, 8.0-μm pore size for 24-well plates) (BD Biosciences, Bedford, MA, USA), grown to confluence and pretreated with mitomycin C (1 μg/mL) over 12 h before an artificial “wound” was created by scratching with a P-200 pipette tip. This point was considered the “0 h,” with the width of the wound photographed under a microscope (×10). The cells were then allowed to close the wound and were photographed at six hours. Wound healing was measured as μm/hour by calculating the reduction in the wound’s width between 0 and 6 h.

### 4.9. D Migration and Invasion Assay

The total of 25 × 10^3^ cells in 250 μL serum-free medium were placed in the upper chambers of 24-well Transwell plates containing a cell culture inserted with 8μm pore size and Matrigel^®^ (invasion, BD BioCoat Matrigel^®^ Invasion Chamber, BD Biosciences, Bedford, MA, USA) or not (3D migration). The lower chamber was filled with 700 μL complete media. After 48 h of incubation, the non-migrating cells on the top were removed using cotton swabs and the migrating cells on bottom surface of the filter were stained using crystal violet. The quantification of the migration and invasion assay was performed by counting the cells on the filter’s lower surface under an inverted microscope at ×20 magnification. The assays were performed in triplicate, with data presented as the average number of invading cells.

### 4.10. Statistical Analysis

For the cells, the data were expressed as the means ± SEM of experiments, independently conducted three times. All statistical analyses were performed using GraphPad Prism 4, a GraphPad Software program (San Diego, CA, USA). The data were analyzed using a two-tailed Student’s t-test for single comparison or one-way ANOVA followed by Dunnett’s multiple comparison analysis. The differences were considered significant at the *p* < 0.05 level.

For YAP staining in samples from the Bio-IFCT0002, the characteristics of patients with (*n* = 362) and without (*n* = 130) YAP IHC were compared using chi-squared tests or Fisher’s exact test for qualitative variables, with Student’s *t*-tests applied for quantitative variables. The associations between the YAP expression or subcellular localization and clinical characteristics were evaluated using Chi-squared, Fisher’s exact or Student’s *t*-tests.

The prognosis values for disease-free survival (DFS) and OS based on IHC scores and subcellular localization were assessed using Cox models. The interaction tests were employed to evaluate predictive values. The IHC scores were first studied as continuous variables ranging from 0 to 300. The median follow-up was estimated using the reverse Kaplan–Meier method. The multivariate Cox models were used to adjust for the patients’ characteristics associated with the corresponding outcomes (DFS, or OS) at *p* < 0.20 in the univariate analysis. The IHC score was dichotomized (negative/positive) as indicated by a fractional polynomial analysis, and the median value selected in YAP analyses by this methodology. A two-step bootstrap re-sampling analysis was performed in order to validate the prognostic model [75]. The data were analyzed using SPSS for Windows Evaluation Version 15.0 (SPSS, Inc., Chicago, IL, USA 2006), the multivariable fractional polynomials (MFP) package of R software (R package Version 1.4.0, original by Gareth Ambler and modified by Axel Benner, 2007) and SAS 9.3 (SAS Institute Inc., Cary, NC, USA).

## 5. Conclusions

Despite advances in surgical techniques, radiotherapy, chemotherapy and most recently, immunotherapy using immune checkpoint inhibitors, NSCLC is still one of the primary causes of cancer-related deaths worldwide [76]. The intrinsic or acquired cellular adaptations to therapy are major factors contributing to treatment failures [77]; identifying these molecular signaling mechanisms appears critical for identifying druggable targets that may either be predictors of therapeutic response or mediators of resistance. Here, this study showed, for the first time, that RASSF1A depletion reduces the ability of bronchial epithelial or lung cancer cells to undergo apoptosis via increasing IAP-2 protein expression and reducing apoptosis-related proteins. The authors have likewise demonstrated that the increase in IAP-2 content in the RASSF1A-depleted HBEC cells is exacerbated by gemcitabine, the administration of which should thus be less efficient to patients with NSCLC exhibiting *RASSF1A* promoter gene methylation. Finally, this study introduces IAP-2 as a critical mediator for the cell consequences of RASSF1A’s loss of expression, which is found in the tumors of up to 30% of NSCLC patients. Whether IAP-2 could represent a new putative target for cytotoxic drugs aiming at specifically treating NSCLC patients with RASSF1A promoter methylation tumors remains to be established. Indeed, emerging evidence derived from recent clinical and experimental studies [78,79] highlights the relevance of pharmacologic inhibitors of IAPs as potential therapeutic agents for cancer therapy. Our results suggest that chemotherapy combined with IAP inhibitors are likely to show elective efficacy in patients with an inactive tumor suppressor gene *RASSF1A*.

## Figures and Tables

**Figure 1 cancers-11-01835-f001:**
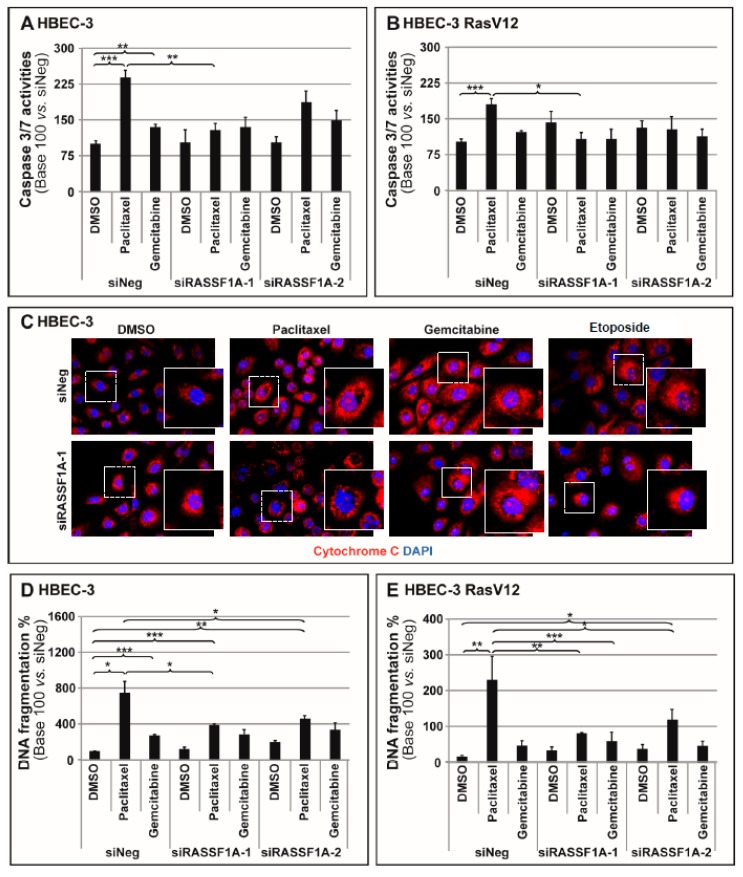
RASSF1A depletion suppresses cell sensitivity to drug-induced apoptosis. HBEC-3 cells were transfected with siNeg or siRASSF1A. The 24-h post-transfection cells were treated for a further 24 h with paclitaxel (10 nM) or gemcitabine (250 nM). (**A**,**B**) The effect of RASSF1A depletion on caspase-3/7 activity was measured by Caspase-Glo^®^ 3/7 Assay kit in (**A**) HBEC-3 and (**B**) HBEC-RasV12 cells undergoing apoptosis using paclitaxel or gemcitabine treatment. (**C**) The effects of RASSF1A depletion on cytochrome C expression were observed by immunofluorescence in HBEC-3 cells undergoing apoptosis induced by paclitaxel or gemcitabine treatment. Magnification: objective ×60. (**D**,**E**) The effects of RASSF1A depletion on DNA fragmentation were measured in (**D**) HBEC-3 and (**E**) HBEC-RasV12 cells undergoing apoptosis induced by paclitaxel or gemcitabine treatment. The data are expressed as the mean ± SEM from three individual experiments. The statistical significance was determined by a Student’s *t*–test: * *p* < 0.05; ** *p* < 0.01; *** *p* < 0.001.

**Figure 2 cancers-11-01835-f002:**
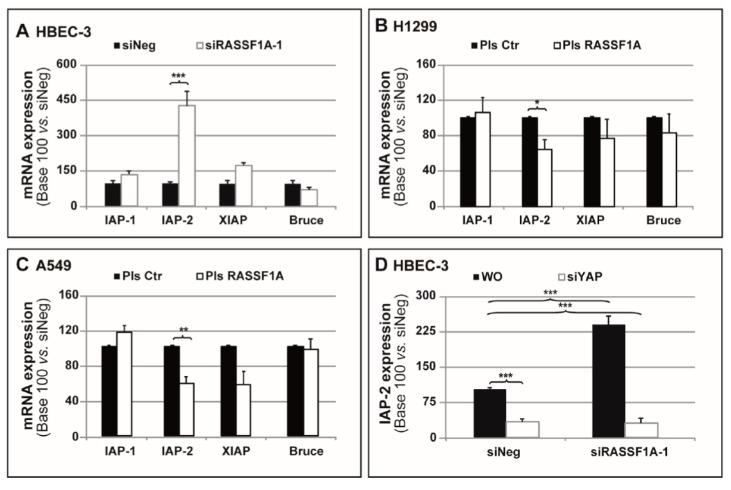
RASSF1A modulates IAP-2 expression. HBEC-3 cells were transiently transfected with siNeg and siRASSF1A-1, whereas A549 and H1299 cells were transfected with plasmid coding wild-type RASSF1A. (**A**–**C**) mRNA expression of IAP-1, IAP-2, surviving, and Bruce was examined by RT-PCR in (**A**) HBEC-3, (**B**) H1299, and (**C**) A549. S16 was used as an internal control. (**D**) mRNA expression of IAP-2 was examined using RT-PCR in HBEC-3 cells that were transiently transfected with siNeg and siRASSF1A-1 both in combination with and without siYAP. S16 was used as an internal control. The data are expressed as the mean ± SEM from three individual experiments. The statistical significance was determined by a Student’s *t*-test. * *p* < 0.05; ** *p* < 0.01; *** *p* < 0.001.

**Figure 3 cancers-11-01835-f003:**
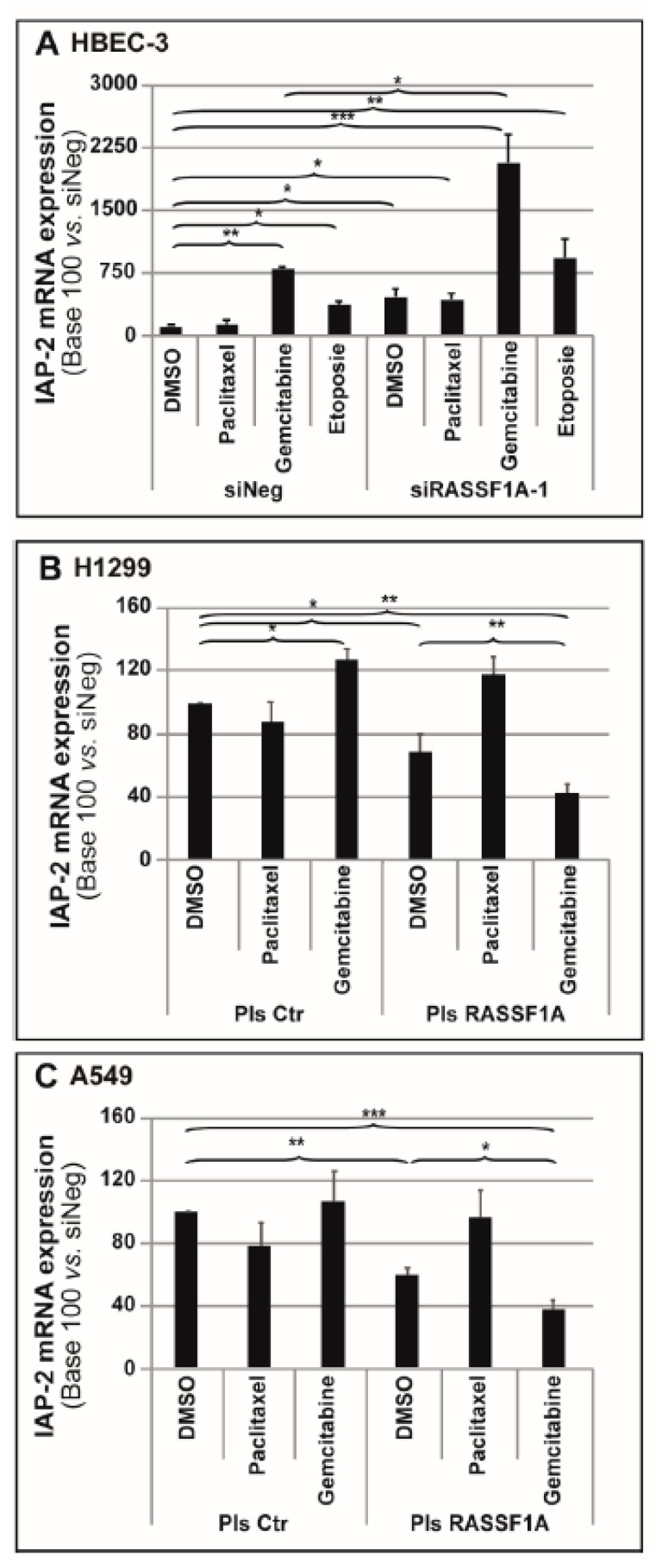
Gemcitabine treatment coincides with strong IAP-2 expression following RASSF1A depletion. HBEC-3 cells were transiently transfected with siNeg and siRASSF1A-1, whereas A549 and H1299 cells were transfected with plasmid coding wild-type RASSF1A. When indicated, 24-h post-transfection cells were treated for a further 24 h with paclitaxel (10 nM), gemcitabine (250 nM), or etoposide (50 µM). (**A**–**C**) mRNA expression of IAP-2 was examined using RT-PCR in (**A**) HBEC-3, (**B**) H1299, and (**C**) A549. S16 was employed as internal control. The data are expressed as the mean ± SEM from three individual experiments. The statistical significance was determined using a Student’s *t*-test. * *p* < 0.05; ** *p* < 0.01; *** *p* < 0.001.

**Figure 4 cancers-11-01835-f004:**
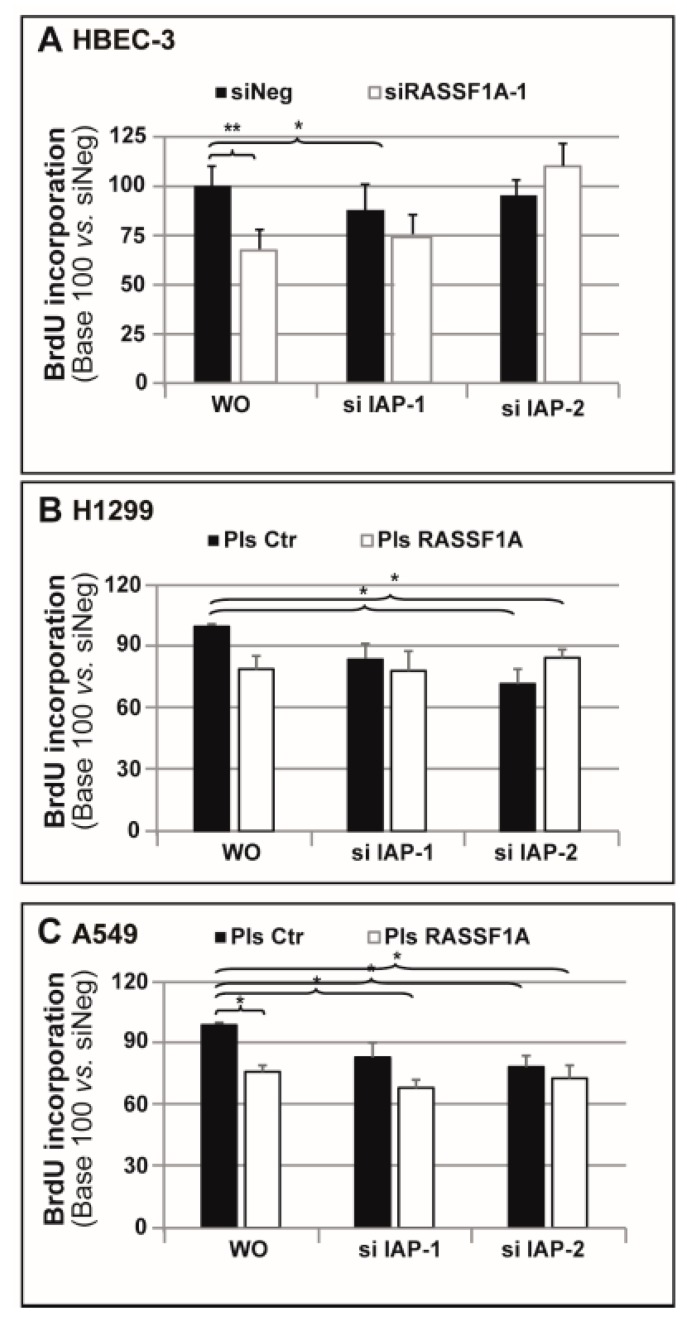
IAP-2 interferes with RASSF1A-mediated cell proliferation. HBEC-3 cells were transiently transfected with siNeg and siRASSF1A-1, whereas A549 and H1299 cells were transfected with plasmid coding wild-type RASSF1A. (**A**–**C**) Cell proliferation was assessed using BrdU incorporation and subsequent spectrophotometric detection at 450 nm wavelength in (**A**) HBEC-3, (**B**) H1299, and (**C**) A549. The data are expressed as the mean ± SEM from three individual experiments. The statistical significance was determined using a Student’s *t*-test. * *p* < 0.05; ** *p* < 0.01.

**Figure 5 cancers-11-01835-f005:**
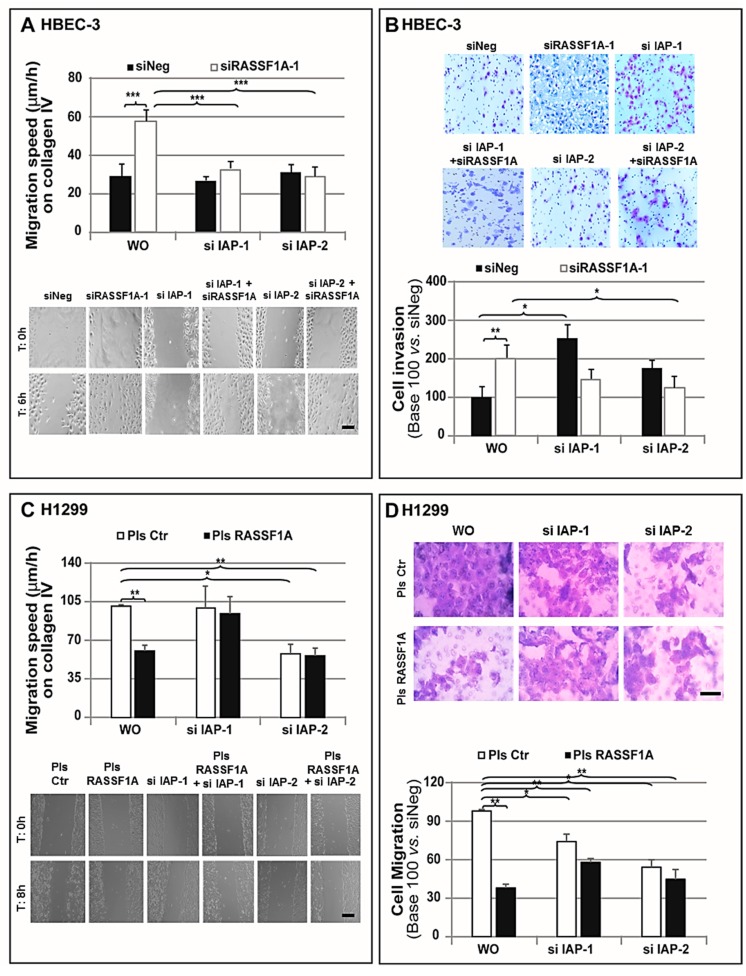
The increase of IAP-2 expression is critical for RASSF1A-mediated effects on cell migration and invasion. HBEC-3 cells were transiently transfected with siNeg and siRASSF1A-1, whereas A549 and H1299 cells were transfected with plasmid coding wild-type RASSF1A both in and without combination with siRNA targeting IAP-1 and IAP-2, (as indicated on x-axis). (**A**,**C**) Migration speed (μm/h) was assessed by the wound-healing assay in (**A**) HBEC-3 (scale bar, 100 μm) and (**C**) H1299 cells (Scale bar, 200 μm). (**B**) Invasion capacity of transfected HBEC-3 cells was measured using Matrigel^®^-coated Invasion Transwell. Relative invasion normalized to that of the cells transfected with siNeg. Scale bar: 50 μm. (**D**) 3D Migration capacity was measured using Transwell without any coating. Relative invasion normalized to that of the cells transfected with control mimic plasmid. Scale bar: 50 μm. The data were expressed as the mean ± SEM from three individual experiments. The statistical significance was determined using a Student’s *t*-test. * *p* < 0.05; ** *p* < 0.01; *** *p* < 0.001.

**Figure 6 cancers-11-01835-f006:**
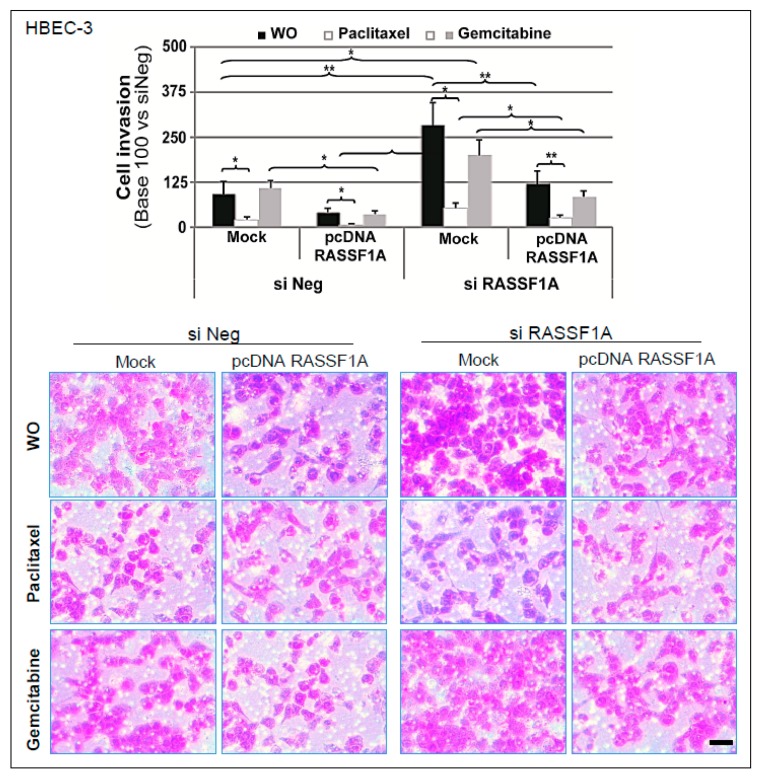
Paclitaxel treatment rescues normal invasion following RASSF1A knockdown. HBEC-3 cells were transiently transfected with siNeg, siRASSF1A, or plasmid coding wild-type RASSF1A. When indicated, 24-h post-transfection cells were treated for a further 24 h with paclitaxel (10 nM) or gemcitabine (250 nM). The invasion capacity of transfected HBEC-3 cells was measured using Matrigel^®^-coated Invasion Transwell. Relative invasion normalized to that of the cells transfected with siNeg. Scale bar: 50 μm. The data are expressed as the mean ± SEM from three individual experiments. The statistical significance was determined using a Student’s *t*-test. * *p* < 0.05; ** *p* < 0.01.

**Table 1 cancers-11-01835-t001:** Correlation of YAP expression with the responses to chemotherapy of 363/528 non-small-cell lung cancer (NSCLC) patients from the IFCT-0002 trial.

YAP Intensity	Response		Total
	Complete or Partial	In Progress or Non-Evaluable	
0 or 1	*n* = 71	*n* = 63	*n* = 134
	(19.94%)	(17.70%)	(37.64%)
2 or 3	*n* = 87	*n* = 135	*n* = 222
	(24.44%)	(37.92%)	(62.36%)
Total	*n* = 158	*n* = 198	*n* = 356
	(44.38%)	(55.62%)	(100%)
	*p* value (Chi-squared association) = 0.011		

**Table 2 cancers-11-01835-t002:** Primers used for RT-PCR in this work.

Target	Primers (5′ → 3′)
RASSF1A	Forward (F): GGG GTC GTC CGC AAA GGC C
	Reverse (R): GGG TGG CTT CTT GCT GGA GGG
IAP-1	F: CCT GGA TAG TCT ACT AAC TGC CT
	R: GCT TCT TGC AGA GAG TTT CTG AA
IAP-2	F: CAG ATT TGG CAA GAG CTG GT
	R: ATT CGA GCT GCA TGT GTC T
Actin	F: CAA CCG TGA AAA GAT GAC CCA G
	R: ATG GGC ACAGTG TGG GTG AC

## Data Availability

All clinical data and full statistical analyses are stored at the IFCT headquarters and can be made available upon request.

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
