# Peer review of "Role of the YAP-1 Transcriptional Target cIAP2 in the Differential Susceptibility to Chemotherapy of Non-Small-Cell Lung Cancer (NSCLC) Patients with Tumor RASSF1A Gene Methylation from the Phase 3 IFCT-0002 Trial"

_cancers, 2019, doi:10.3390/cancers11121835_

Round 1

Reviewer 1 Report

In this study, the authors address that gene silencing (mimics promoter gene methylation) of RASSF1A can reduce the apoptosis as well as increase the proliferation, migration and invasion in the immortalized human bronchial epithelial cell lines or non-small cell lung cancer (NSCLC) cell lines, through upregulation of IAP-2 expression and suppression of apoptosis-related proteins. In addition, the authors explore the roles of RASSF1A and IAPs to affect efficiency of paclitaxel or gemcitabine in NSCLC patients' chemotherapy. The evidences are very complete. However, some questions need to clarify.

We don't observe any information about the method of plasmids encoding wild type RASSF1A (pcDNA3-RASSF1A) and control mimic in the Reference 4.

How to explain that HBEC-3 cell proliferation felt down significantly after RASSF1A depletion, and concomitant depletion of RASSF1A and IAP-2 rescued RASSF1A-depleted cells defect of proliferation in Figure 4A.

What is the meanings which authors want to express in this sentence "According, the H1299 and A549 cells, where IAP-1 RNAi had no effect, the migration speed after IAP-2 knockdown, either in presence or absence of RASSF1A, was comparable to the level of control cells transfected with RASSF1A plasmid."

How to count the cell numbers which have migrated or penetrate in 3D migration and invasion assays?

We observe that the suppression of cell invasion in siIAP-1+ siRASSF1A group was increased more than that in in siIAP-2+ siRASSF1A group in upper panel of Figure 5B. It seems to not match with the bar-scale figure (the lower panel of Figure 5B).

Author Response

In this study, the authors address that gene silencing (mimics promoter gene methylation) of RASSF1A can reduce the apoptosis as well as increase the proliferation, migration and invasion in the immortalized human bronchial epithelial cell lines or non-small cell lung cancer (NSCLC) cell lines, through upregulation of IAP-2 expression and suppression of apoptosis-related proteins. In addition, the authors explore the roles of RASSF1A and IAPs to affect efficiency of paclitaxel or gemcitabine in NSCLC patients' chemotherapy. The evidences are very complete.
We wish to thank Reviewer #1 for his/her enthusiastic support of our manuscript. We very much appreciate such positive comments.
However, some questions need to clarify.
1) We don't observe any information about the method of plasmids encoding wild type RASSF1A (pcDNA3-RASSF1A) and control mimic in the Reference 4.
In reply to Reviewer #1’s concern, please note that the information concerning siRNA, plasmid DNA, or control mimics, has now been further detailed in the Supplementary Tables S2 and S3, as well as in reference [4].
To make it easier for readers to find this information, we have changed the sentence referring to it as follows: “Plasmids encoding wild-type RASSF1A (pcDNA3-RASSF1A) and control mimic (Addgene, Cambridge, MA, USA) have been described previously in the supplementary data section [4]. “

2) How to explain that HBEC-3 cell proliferation felt down significantly after RASSF1A depletion, and concomitant depletion of RASSF1A and IAP-2 rescued RASSF1A-depleted cells defect of proliferation in Figure 4A.
As we did not demonstrate this in our research, the explanation remains speculative and we thus decided to not expand on this point in the manuscript’s Discussion section.
As explained in the Introduction section, IAPs bind to and inhibit caspase enzymes, which are the terminal effectors of apoptosis. However, short prodomain caspases, particularly caspase-3, were shown to additionally exert several non-apoptotic functions, especially concerning the modulation of cell growth, as described in the nice review by Lamkanfi et al., 2006 (https://www.nature.com/articles/4402047), in which it is stated that “For example, nuclei of dividing cells in the proliferative regions of rat forebrain display the presence of active caspase-3. In proliferating lymphoid cells, caspase-mediated cleavage of the cyclin-dependent kinase (CDK) inhibitor p27KIP1 contributes to the induction of cell cycle progression.58 In contrast, hyperproliferation of B cells is observed in caspase-3 knockout mice, indicating that caspase-3 may act as a negative regulator of B-cell cycling. Although the CDK inhibitor p21 is a known inhibitor of cell cycle progression, p21 can also promote cell proliferation when associated with PCNA, a processivity factor which promotes entry into mitosis. In line with these observations, caspase-3-mediated cleavage of p21 at the C terminal PCNA-binding site specifically abolishes interaction of p21 with PCNA, thus explaining the anti-proliferative effect of this cleavage event in B cells.”
With this background in mind, we hypothesize that inhibition of IAP2 may enable caspase 3 to fulfill its regulator function on cell cycle progression, which explains why its inhibition allows RASSF1A-depleted cells to regain a proliferation comparable to that observed under control conditions (siNeg).

3) What is the meanings which authors want to express in this sentence "According, the H1299 and A549 cells, where IAP-1 RNAi had no effect, the migration speed after IAP-2 knockdown, either in presence or absence of RASSF1A, was comparable to the level of control cells transfected with RASSF1A plasmid."
We are sorry that this sentence that was not very clear. It has been modified and reformulated as follows:
“Accordingly, IAP-1 silencing did not affect the migration velocity of H1299 and A549 cells either expressing or not RASSF1A, while IAP-2 silencing decreased significantly such migration velocity of H1299 and A549 cells either expressing or not RASSF1A (Figure 5C & S5A).”

4) How to count the cell numbers which have migrated or penetrate in 3D migration and invasion assays?
Quantification of the migration and invasion assay was performed by counting the cells on the filter’s lower surface under an inverted microscope at x20 magnification.
This information has now been added to the Material and Method section.

5) We observe that the suppression of cell invasion in siIAP-1+ siRASSF1A group was increased more than that in in siIAP-2+ siRASSF1A group in upper panel of Figure 5B. It seems to not match with the bar-scale figure (the lower panel of Figure 5B).
We are sorry for this discrepancy, which is actually due to a wrong choice among representative photos. The photo of the condition "siIAP1 + siRASSF1A" has now been removed. To better understand the histogram, please consult the new panel B of Figure 5.

Reviewer 2 Report

This article is centred on the study of Non-Small Cell Lung Cancer (NSCLC) cells that present RASSF1A gene methylation, or not, and their response to the chemotherapy agents paclitaxel and gemcitabine. The interest of the study is that early stage NSCLC patients that present RASSF1A gene methylation presented a better response to paclitaxel than to gemcitabine treatment in a previous phase 3 trial. The authors study two cell lines that express RASSF1A and other two that do not express the gene due to promoter methylation. RASSF1A gene expression is either silenced or induced in these cells and the effects of paclitaxel and gemcitabine determined. In the course of these studies the authors identify IAP-2 as an important protein involved in the pathogenic mechanism of tumorigenesis induced by RASSF1A silencing and as a novel biomarker in lung cancer.

The study presented is extensive and the results obtained interesting in the cancer and NSCLC fields. The results are sound and support the conclusions of the article. There are, however, some points that should be addressed by the authors as follows:

The study presented is rather complex, with several proteins and cellular pathways involved so that it is difficult to follow at some points. One final scheme summarizing the results obtained and the proposed mechanisms of paclitaxel sensitivity and gemcitabine resistance would greatly help the non-specialized reader to better understand the article.

The statistical significance of the data shown in Table 2 is difficult to understand. In the text (lines 192-193) the authors compare the percentage of patients with negative or low YAP expression and poor response (17.70%) with those with strong immunostaining of YAP and poor response (37.92%). However these percentages are not referred to the total of the patients of each group. For example, the patients with YAP intensity 0 or 1 (134) represent 37.64% of the total population and among them 63 (47%) do not respond to the treatment, representing the 17.70% of the total population. In the case of tumors with 2 or 3 YAP intensity, 135 out of 222 patients (60.8%) did not show response to treatment. I consider that the authors compare these two percentages of non-response patients: 47% versus 60.8% but this is not clear in the text. It rather seems that the comparison is made with the percentages of the total population: 17.70 vs 37.64 % when the number of patients in the groups of low and high intensity is not the same.

Lines 288-291. The authors state in this paragraph that “combined RASSF1A and IAP-2 depletion dramatically blocked the siRASSF1A-induced invasion (Figure 5B)”. However, this change is not observed in Figure 5B and combined depletion of RASSF1A and IAP-2 seems to have the same effect on cell invasion than combined depletion of RASSF1A and IAP-1.

Lines 329-332. This paragraph is not very clear. The authors indicate that paclitaxel does not interfere with IAP-2 expression and that because of this reason, contributes efficiently to limit cell migration and subsequent metastatic potential. However, it could be interpreted that by non interfering with IAP-2 expression, cell migration would not be increased. Is there any reason to postulated that it would be limited?.

Lines 435-437. This paragraph has been repeated in lines 423-425.

Figure S3 legend. Panel C is not described.

Author Response

This article is centred on the study of Non-Small Cell Lung Cancer (NSCLC) cells that present RASSF1A gene methylation, or not, and their response to the chemotherapy agents paclitaxel and gemcitabine. The interest of the study is that early stage NSCLC patients that present RASSF1A gene methylation presented a better response to paclitaxel than to gemcitabine treatment in a previous phase 3 trial. The authors study two cell lines that express RASSF1A and other two that do not express the gene due to promoter methylation. RASSF1A gene expression is either silenced or induced in these cells and the effects of paclitaxel and gemcitabine determined. In the course of these studies the authors identify IAP-2 as an important protein involved in the pathogenic mechanism of tumorigenesis induced by RASSF1A silencing and as a novel biomarker in lung cancer.
The study presented is extensive and the results obtained interesting in the cancer and NSCLC fields. The results are sound and support the conclusions of the article.
First, we would like to express our thanks to Reviewer #2 for his/her enthusiastic support of our manuscript.
There are, however, some points that should be addressed by the authors as follows:
1. The study presented is rather complex, with several proteins and cellular pathways involved so that it is difficult to follow at some points. One final scheme summarizing the results obtained and the proposed mechanisms of paclitaxel sensitivity and gemcitabine resistance would greatly help the non-specialized reader to better understand the article.
We fully agree with Reviewer #2’s comment and that is why we made a graphical abstract, summarizing this paper’s main results. However, based on Reviewer #2’s query, we assume that the graphical abstract we provided has been overlooked.
For this reason, we have now integrated this graphical abstract into our manuscript as its last figure. For further information, please consult the new Figure 7.
2. The statistical significance of the data shown in Table 2 is difficult to understand. In the text (lines 192-193) the authors compare the percentage of patients with negative or low YAP expression and poor response (17.70%) with those with strong immunostaining of YAP and poor response (37.92%). However these percentages are not referred to the total of the patients of each group. For example, the patients with YAP intensity 0 or 1 (134) represent 37.64% of the total population and among them 63 (47%) do not respond to the treatment, representing the 17.70% of the total population. In the case of tumors with 2 or 3 YAP intensity, 135 out of 222 patients (60.8%) did not show response to treatment. I consider that the authors compare these two percentages of non-response patients: 47% versus 60.8% but this is not clear in the text. It rather seems that the comparison is made with the percentages of the total population: 17.70 vs 37.64 % when the number of patients in the groups of low and high intensity is not the same.
The statistical analysis was performed taking into account the patient percentages of each group in relation to the size of the entire cohort (n = 356).
To facilitate its understanding, we have now reformulated this paragraph in the new manuscript, which now reads as follows:
“Overall, 63 of the 356 (17.7%) patients studied in this trial showed no response and did not express or expressed low (0-1 intensities) tumoral YAP intensity (i.e., 37.6% [63/134] of patients with low YAP intensity tumor), whereas 135 patients showed no response and a strongly expressed YAP tumor intensity (i.e., 60.81% (135/222) of patients with strong tumoral YAP intensity)”.
3. Lines 288-291. The authors state in this paragraph that “combined RASSF1A and IAP-2 depletion dramatically blocked the siRASSF1A-induced invasion (Figure 5B)”. However, this change is not observed in Figure 5B and combined depletion of RASSF1A and IAP-2 seems to have the same effect on cell invasion than combined depletion of RASSF1A and IAP-1.
We fully agree with Reviewer #2’s comment in that the profiles "siRASSF1A + siIAP1" and "siRASSF1A + siIAP2" are comparable. However, there is no statistical difference between the histograms "siRASSF1A" versus "siRASSF1A + siIAP1," whereas the statistical analysis results in a p <0.05 when the histograms "siRASSF1A" versus "siRASSF1A + siIAP2" are being compared.
4. Lines 329-332. This paragraph is not very clear. The authors indicate that paclitaxel does not interfere with IAP-2 expression and that because of this reason, contributes efficiently to limit cell migration and subsequent metastatic potential. However, it could be interpreted that by non interfering with IAP-2 expression, cell migration would not be increased. Is there any reason to postulate that it would be limited?
Indeed, we believe that paclitaxel limits the cell migration in different ways: 1) by not inducing IAP-2 expression; 2) by stabilizing microtubules as discussed in the Discussion section’s last paragraph of the new manuscript.
5. Lines 435-437. This paragraph has been repeated in lines 423-425.
We apologize for this duplication, and the previous lines 435-437 have now been removed.
6. Figure S3 legend. Panel C is not described.
This panel, which was previously unclear, has now been further specified: “Figure S3: Relationship between Yap cytoplasmic or nuclear expression and the rates of overall survival.
A) Representative intensity of YAP expression measured by IHC (example of score assignment: negative (I=0), weak (I=1), moderate (I=2) and strong (I=3)). B) Flowchart of patient selection and inclusion C-D) Survival analyses in relation to intensity (C) or nuclear/cytosolic (D) YAP expression in NSCLC. E) Survival analysis in NSCLC patients in relation to YAP expression using the Cancer Genome Atlas cohort.”

Finally, we want to underline that the manuscript’s English has now been thoroughly corrected by using a professional English editing service (Cremer consulting SARL: https://cremerconsulting.com/en/)

Round 2

Reviewer 1 Report

All the responses are reasonable and acceptable.